# Decreased lung function is associated with elevated ferritin but not iron or transferrin saturation in 42,927 healthy Korean men: A cross-sectional study

Jonghoo Lee[1☯], Hye kyeong Park[2☯], Min-Jung Kwon[3], Soo-Youn Ham[4], Joon Mo Kim[5], Si-Young Lim[6], Jae-Uk Song[6]*

1 Department of Internal Medicine, Jeju National University Hospital, Jeju National University School of Medicine, Jeju, Republic of Korea, 2 Division of Pulmonary and Critical Care Medicine, Department of Internal Medicine, Ilsan Paik Hospital, Inje University College of Medicine, Ilsan, Republic of Korea, 3 Department of Laboratory Medicine, Kangbuk Samsung Hospital, Sungkyunkwan University School of Medicine, Seoul, Republic of Korea, 4 Department of Radiology, Kangbuk Samsung Hospital, Sungkyunkwan University School of Medicine, Seoul, Republic of Korea, 5 Department of Ophthalmology, Kangbuk Samsung Hospital, Sungkyunkwan University School of Medicine, Seoul, Republic of Korea, 6 Division of Pulmonary and Critical Care Medicine, Department of Internal Medicine, Kangbuk Samsung Hospital, Sungkyunkwan University School of Medicine, Seoul, Republic of Korea

☯ These authors contributed equally to this work.
* khfever76@gmail.com

**Data Availability Statement:** Due to ethical restrictions imposed by the Institutional Review Board of Kangbuk Samsung Hospital, the patient

# Abstract

## Objectives

Though elevated ferritin level and decreased lung function both predispose people to cardio-metabolic disease, few reports have investigated the association between them. Furthermore, it remains unclear whether the association reflects a change in iron stores or an epiphenomenon reflecting metabolic stress. Therefore, we looked for possible associations between ferritin, iron, and transferrin saturation (TSAT) and lung function to clarify the role of iron-related parameters in healthy men.

## Methods

We conducted a cohort study of 42,927 healthy Korean men (mean age: 38.6 years). Percent predicted forced expiratory volume in one second (FEV1%) and forced vital capacity (FVC%) were categorized into quartiles. Adjusted odds ratios (aORs) and 95% confidence intervals (using the highest quartile as reference) were calculated for hyperferritinemia, high iron, and high TSAT after controlling for potential confounders.

## Results

The median ferritin level was 199.8 (141.5–275.6) ng/mL. The prevalence of hyperferritinemia (defined as >300 ng/mL) was 19.3%. Subjects with hyperferritinemia had lower FEV1% and FVC% than those with normal ferritin level with a slight difference, but those were statistically significant (99.22% vs.99.61% for FEV1%, p = 0.015 and 98.43% vs. 98.87% for

data are not available for distribution outside of the Kangbuk Samsung Hospital. For additional information, please contact JiinAhn (Jiin57. ahn@samsung.com) or the Institutional Review Board of Kangbuk Samsung Hospital (contact information below). - Address: 29 Saemunan-ro, Jongno-Gu, Seoul, Korea (03181) - E-mail: irb. kbsmc@samsung.com - Telephone: 82-2-2001-1943, 1945 - Fax: 82-2-2001-1946

**Funding:** The 2020 scientific promotion program funded by Jeju National University.

**Competing interests:** NO authors have competing interests.

FVC, p = 0.001). However, FEV1/FVC ratio was not significantly different between groups (P = 0.797). Compared with the highest quartile, the aORs for hyperferritinemia across decreasing quartiles were 1.081 (1.005–1.163), 1.100 (1.007–1.200), and 1.140 (1.053–1.233) for FEV1% (p for trend = 0.007) and 1.094 (1.018–1.176), 1.101 (1.021–1.188), and 1.150 (1.056–1.252) for FVC% (p for trend = 0.001). However, neither FEV1% nor FVC% was associated with iron or TSAT.

## Conclusions

Hyperferritinemia was associated with decreased lung function in healthy Korean men, but iron and TSAT were not. Longitudinal follow-up studies are required to validate our findings.

## Introduction

Iron is essential for multiple metabolic processes, but it is hazardous in excess amounts because its ability to catalyze the generation of reactive oxygen species can cause oxidative stress and damage cellular membranes [1]. Ferritin is a specialized iron storage protein that regulates body iron homeostasis and reflects iron stores in the body. Ferritin also can become elevated in the presence of oxidative stress and inflammation irrespective of iron status and can contribute to various clinical diseases, especially pulmonary and cardio-metabolic diseases [2,3]. Moreover, decreased lung function is associated with oxidative stress and systemic inflammation [4] and increased risk of insulin resistance, diabetes, and cardiovascular disease [5]. This suggests that decreased lung function could be associated with elevated serum ferritin level in pathological conditions.

Even in healthy subjects, airway epithelial cells could be exposed to oxidative stress and inflammation because of ambient air pollution aerosols, which recently have increased rapidly as a consequence of regional urbanization and industrialization [6]. Air pollution particulate matter (PM) contains transition metals such as iron (usually most abundant), which can be mobilized from the PM to lung epithelial cells and disrupt iron homeostasis both in the lung and systemically [7,8]. Iron overabundance relative to metabolic needs inside lung epithelial cells can result in ferritin release from damaged cells, which could result in elevated serum ferritin concentration and loss of lung function under normal physiological conditions [6–8].

Nevertheless, epidemiological evidence to support that hypothesis is scarce. To date, only four reports have investigated a potential association between ferritin and lung function [9–12]; two of them found a positive association [9,10], and the other two found no association [11,12]. However, these previous studies were not conducted only in healthy subjects. Recent studies have continuously shown that pulmonary [2] and cardio-metabolic diseases [13–18] are a recognized complication of excess iron accumulation, and such patients are prone to poor lung function [2,5]. This suggests that inclusion of patients with clinical disease could distort the magnitude of association between lung function and ferritin. Thus, the exact nature of the relationship between ferritin and lung function, if one exists, remains unclear. It also remains unclear whether the previous results reflect a relationship between lung function and a recognized marker of iron status or an epiphenomenon in which an increased ferritin level reflects overall inflammation. Therefore, we investigated the relationships between lung function and ferritin and lung function and other biomarkers of iron metabolism, including transferrin saturation (TSAT), in healthy Korean men.

## Materials and methods

### Study design and population

This study began with data from 189,154 individuals who underwent a comprehensive health examination at Kangbuk Samsung Hospital Health Screening Centers in 2015. Initially, we extracted 188,596 participants with a recorded serum ferritin level and spirometry data. We excluded 85,455 participants who had a ventilation disorder (pure restriction: forced expiratory volume in one second (FEV1) to forced vital capacity (FVC) [FEV1/FVC] $\geq$ 0.7 and FVC < 80% predicted; pure obstruction: FEV1/FVC < 0.7 and FVC $\geq$80% predicted and mixed ventilation disorder: FEV1/FVC < 0.7 and FVC < 80% predicted) [19] and self-reported history of disease or were currently receiving medication for any disease to 1) test our hypothesis with robustness, 2) yield more meaningful results, and 3) minimize potential confounders. Specific details of comorbidities were unavailable because the medical history questionnaire required only yes/no responses. We also excluded 11,942 subjects with missing medical history data or data about smoking habits or alcohol consumption. Because some subjects had more than more than one exclusion criterion, 91,199 subjects were ultimately eligible for initial inclusion in the study. The reference range for serum ferritin is usually from 30 to 300 ng/mL in men and from 15 to 200 ng/mL in women. These ranges seem to vary across reference populations and according to age and sex, which are the most important determinants of serum ferritin [20]. There has been no consensus on criteria for hyperferritinemia. Our cohort consisted of mostly middle aged Asians with higher socioeconomic and education levels and urban habitation with the highest annual PM concentrations in the world. Thus, these characteristics can contribute to more increase in the amount of mobilized iron especially in men than women in our cohort [8,21–26]. The large difference in serum ferritin level between men and women can contribute to much higher lifetime iron stores in men than in women. Therefore, the prevalence of hyperferrtinemia can be markedly different between men and women. Accordingly, we excluded women due to the small proportion (451, 0.9%) with hyperferritinemia, defined as > 200 ng/mL, leaving too few women for us to obtain statistically valid results. Thus, we decided to analyze the data from 42,927 men (Fig 1)

The study was approved by the Institutional Review Board of Kangbuk Samsung Hospital, which waived the requirement for informed consent because we retrospectively accessed a de-identified database for our analyses.

### Anthropometric and laboratory measurements

Data on demographic characteristics, history of smoking and alcohol use, medical history, current regular use of medications, and any clinical symptoms were collected through a self-administered questionnaire. Smoking habits were classified as follows: non-smokers, ex-smokers (no current smoking but regular smoking in the past), and current smokers (at least one cigarette per day). Alcohol history was considered positive if the subject had used alcohol in the past, even if they had stopped drinking.

Physical characteristics and serum biochemical parameters were measured by trained nurses, as previously reported [27,28]. Obesity was defined as Body mass index (BMI) $\geq$25 kg/m$^2$ [29]. Insulin resistance was assessed using the homeostasis model assessment of insulin resistance (HOMA-IR) equation: fasting blood insulin ($\mu$U/ml) $\times$ fasting blood glucose (mmol/l)/22.5 [30]. Serum iron concentration was measured using an automatic chemistry analyzer (Cobas 8000 c702; Roche Diagnostics, Tokyo, Japan) and a colorimetric assay. The serum ferritin was determined using an electrochemiluminescence immunoassay (Cobas 8000 e602; Roche Diagnostics) based on the sandwich principle. We defined subjects with

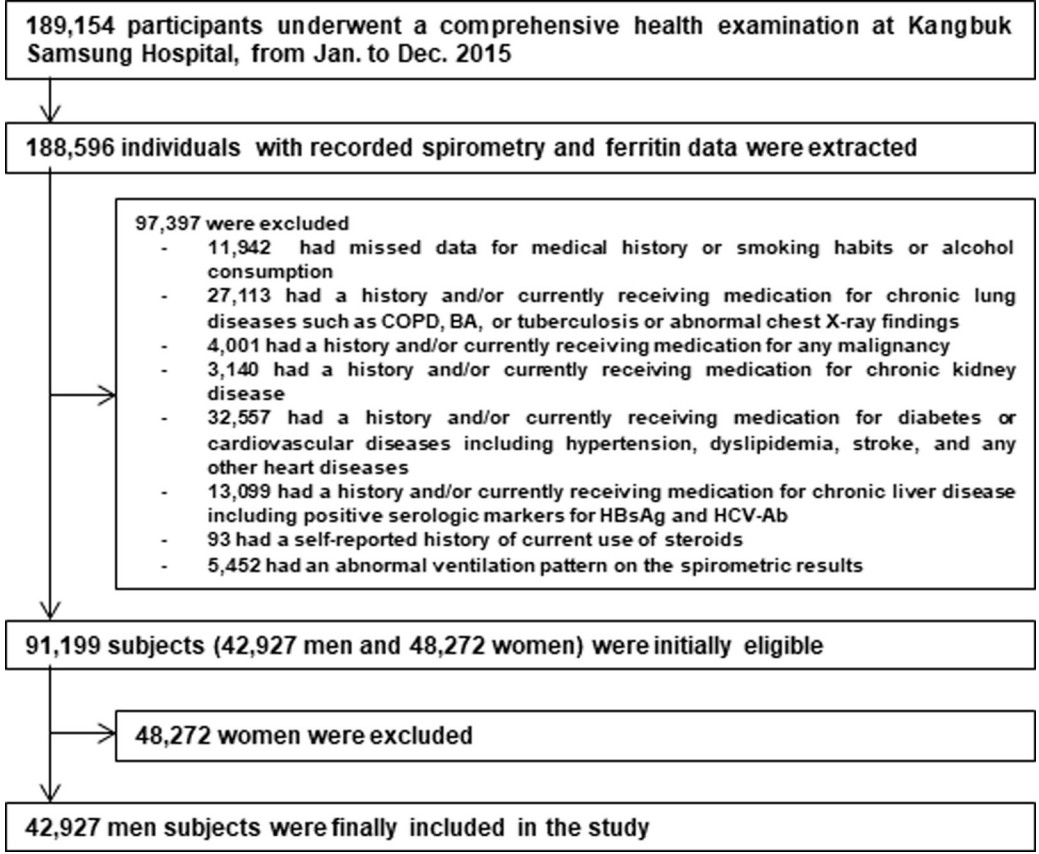

**Fig 1. Flow chart of study participants.** BA = bronchial asthma; COPD = chronic obstructive pulmonary disease; HBsAg = hepatitis B virus surface antigen; HCV = hepatitis C virus.

ferritin > 300 ng/mL, TSAT > 50%, and iron > 175 μg/dL as having hyperferritinemia, high TSAT, and high iron, respectively [20,31]. The Laboratory Medicine Department at Kangbuk Samsung Hospital has been accredited and participates annually in inspections and surveys by the Korean Association of Quality Assurance for Clinical Laboratories.

## Lung function measurement

Spirometry was performed as recommended by the American Thoracic Society [32] using the Vmax22 system (Sensor-Medics, Yorba Linda, CA). A bronchodilator was not administered prior to spirometry. The highest forced expiratory volume in 1 s (FEV1) and forced vital capacity (FVC) values from three or more tests with acceptable curves were used for further analyses. The predicted values for FEV1 and FVC were calculated from equations to obtain in a representative Korean population sample [33]. To calculate the predicted FVC% and predicted FEV1%, we divided the measured value (L) by the predicted value (L) and converted the quotient into a percentage. Subjects were divided into quartiles of FVC% and FEV1%.

## Statistical analyses

Continuous variables are described as the mean ± standard deviation or median and interquartile range, and categorical variables are expressed as number and percentage. The normality of variables was assessed with the Kolmogorov-Smirnov test. The baseline data stratified by

upper normal limit values for biomarkers of iron metabolism and quartiles of ventilator function were compared using t-tests or Mann–Whitney U tests for two-group comparisons and one-way analysis of variance or Kruskal-Wallis tests for comparisons among quartiles. Chi-square tests or Fisher's exact tests were used for categorical variables.

To analyze the significance of differences between subjects stratified by upper normal limit values for biomarkers of iron metabolism, all covariates were transformed into categorical variables: high or low and with or without. Chi-square tests or Fisher's exact tests were used to assess the significance of differences between dichotomous variables.

Binary logistic regression was used to assess the association between FVC% and FEV1% quartiles and above the upper normal limit values for biomarkers of iron metabolism: model 1 was adjusted for age, BMI, and MBP; model 2 was adjusted as in model 1 plus smoking and alcohol; model 3 was adjusted as in model 2 plus variables with a p value<0.05 in the univariate analyses. Because FVC (L) and FEV1 (L) were strongly correlated (r = 0.947, p<0.001), these parameters were assessed separately to avoid confounding effects. The strength of associations was estimated using odds ratios (ORs) and 95% confidence intervals (CIs). All tests were two-sided, and a p value <0.05 was considered statistically significant. Data were analyzed using IBM SPSS Statistics 19.0 (IBM, Armonk, NY, USA).

## Results

Table 1 compares the baseline characteristics of the enrolled subjects between the groups with and without hyperferritinemia. The mean age was 38.6 years. The median ferritin level was 199.3 (141.5–275.5) ng/mL, and the prevalence of hyperferritinemia was 19.3%. Comparison of clinical variables between the two groups showed small, but significant difference in age, smoking habit, alcohol intake, liver function, CRP, blood pressure and a variety of metabolic parameters, including BMI, fasting glucose, and HbA1c. Compared with the normal ferritin group, subjects in the hyperferritinemia group had lower values of spirometry with a narrow margin, although those were statistically significant. However, the difference in FEV1 (L)/FVC (L) between the groups was not statistically significant (p = 0.797). Conventional cardio-metabolic parameters (glucose, HbA1c, total cholesterol, triglycerides, LDL-cholesterol, insulin, HOMA-IR, and hsCRP) were adversely affected in subjects with hyperferritinemia compared with subjects with normal ferritin (Table 2).

A comparison of clinical characteristics between subjects with and without high iron or TSAT is shown in Table 3. Subjects with high iron and TSAT were more likely than others to drink and smoke with a slight difference. However, both high iron and TSAT were inversely associated with hsCRP and metabolic values, including BMI, HbA1c, insulin, and HOMA-IR, although insulin was only related to TSAT. Compared with lowest quartiles of lung function, the prevalence of high iron or TSAT was not significantly different in accordance with an increase in lung function quartile, except for the prevalence of high TSAT according to quartile of FVC%.

We analyzed the associations between lung function and prevalence of hyperferritinemia, high iron, and high TSAT to investigate whether lung function was independently associated with ferritin and other biomarkers of iron metabolism (Table 4 and Fig 2). Compared with the highest quartile (the reference group) of FVC%, the aORs for hyperferritinemia across the decreasing quartile of lung function were 1.094 (1.018–1.176), 1.101 (1.021–1.188), and 1.150 (1.056–1.252), respectively (p for trend = 0.001). Similar results were observed across FEV1% quartiles (p for trend = 0.007). The aORs for high TSAT are lower for subjects with lower FVC % according to model 1 and model 2. However, there was no significant relationship between

**Table 1. Comparison of the demographic and clinical characteristics of the study subjects.**

| | All subjects (n = 42,927) | Normal ferritin (ferritin ≤300 ng/mL) (n = 34,743, 80.7%) | Hyperferritinemia (ferritin >300 ng/mL) (n = 8,184, 19.3%) |
|---|---|---|---|
| Age (years) [*] | 38.6 ± 7.0 | 38.7 ± 7.0 | 38.1 ± 6.7 |
| 19–30 | 5,267 (12.3) | 4,281 (12.3) | 986 (12.0) |
| 31–40 | 20,981 (48.9) | 16,691 (48.0) | 4,290 (52.4) |
| 41–50 | 14,369 (33.5) | 11,821 (34.0) | 2,548 (31.1) |
| 51–60 | 2164 (5.0) | 1,823 (5.2) | 341 (4.2) |
| ≥61 | 146 (0.3) | 127 (0.4) | 19 (0.2) |
| BMI (kg/m$^2$) [*] | 24.3 ± 2.7 | 24.1 ± 2.7 | 25.2 ± 2.9 |
| Smoking status[†] | | | |
| Non-smoker | 16,953 (39.5) | 13,831 (39.8) | 3,122 (38.1) |
| Former smoker | 13,745 (32.0) | 11,031 (31.8) | 2,714 (33.2) |
| Current smoker | 12,229 (28.5) | 9,881 (28.4) | 2,348 (28.7) |
| No alcohol drinking[*] | 5,109 (11.9) | 4,390 (12.6) | 719 (8.8) |
| Total cholesterol (mg/dL)[*] | 194 ± 30 | 193 ± 30 | 200 ± 32 |
| Triglycerides (mg/dl)[*] | 106 (76–150) | 102 (74–143) | 124 (88–178) |
| LDL-C (mg/dl)[*] | 128 ± 29 | 127 ± 28 | 133 ± 30 |
| Total bilirubin (mg/dL) (n = 42,926)[*] | 0.9 ± 0.4 | 0.9 ± 0.4 | 1.0 ± 0.4 |
| ALT (U/L) (n = 42,925)[*] | 21 (18–25) | 20 (17–24) | 23 (19–30) |
| Serum creatinine (mg/dL) | 0.9 ± 0.1 | 0.9 ± 0.1 | 0.9 ± 0.1 |
| Fasting glucose (mg/dl)[*] | 96 ± 11 | 95 ± 9 | 98 ± 16 |
| Insulin (μIU/ml) (n = 42,871)[*] | 5.6 (3.9–8.0) | 5.4 (3.8–7.7) | 6.4 (4.4–9.3) |
| HOMA-IR (n = 42,871)[*] | 1.31 (0.89–1.93) | 1.27 (0.86–1.84) | 1.53 (1.02–2.28) |
| HbA1c (%) (n = 42,921)[*] | 5.5 ± 0.4 | 5.5 ± 0.3 | 5.6 ± 0.5 |
| hsCRP (mg/l) (n = 34,979)[*] | 0.05 (0.03–0.09) | 0.05 (0.03–0.09) | 0.06 (0.03–0.11) |
| SBP (mmHg) (n = 42,924)[*] | 114 ± 11 | 113 ± 10 | 116 ± 11 |
| DBP (mmHg) (n = 42,924)[*] | 73 ± 9 | 73 ± 9 | 75 ± 9 |
| MBP (mmHg) (n = 42,924)[*] | 87 ± 9 | 86 ± 9 | 88 ± 9 |
| Spirometry values | | | |
| FVC (L) [†] | 4.734 ± 0.556 | 4.745 ± 0.545 | 4.730 ± 0.557 |
| FEV1(L) [†] | 3.884 ± 0.474 | 3.891 ± 0.467 | 3.879 ± 0.477 |
| FEV1(L)/FVC(L) ratio | 0.81 ± 0.02 | 0.81 ± 0.02 | 0.81 ± 0.02 |
| FVC% predicted[†] | 98.62 ± 8.83 | 98.87 ± 8.88 | 98.43 ± 8.63 |
| FEV1% predicted[†] | 99.54 ± 9.32 | 99.61 ± 9.36 | 99.22 ± 9.12 |
| Ferritin level (ng/mL)[*] | 199.3 (141.5–275.5) | 176.9 (129.9–227.2) | 368.7 (329.0–434.2) |
| Iron (μg/dL) (n = 35,890) | 129 (104–156) | 129 (104–157) | 129 (105–154) |
| Transferrin saturation (%) (n = 34,873)[†] | 43 (34–53) | 43 (34–53) | 43 (35–53) |

Continuous variables are expressed as mean ± standard deviation or median and interquartile range. Categorical variables are described as number and percentage. We recorded subject numbers with available clinical parameters. Unless otherwise indicated, the available subject number was 42,927.

[*]p<0.001 compared with hyperferritinemia;

[†]p<0.05 compared with hyperferritinemia.

ALT = alanine aminotransferase; BMI = body mass index; DBP = diastolic blood pressure; FVC% predicted = percent predicted forced vital capacity; FEV1% predicted = percent predicted forced expiratory volume in 1s; HbA1c = hemoglobin A1c; HOMA-IR = homeostasis model assessment of insulin resistance; hsCRP = high-sensitivity C-reactive protein; LDL = low-density lipoprotein; MBP = mean blood pressure; SBP = systolic blood pressure.

MBP = diastolic BP + (average systolic BP—average diastolic BP)/3.

**Table 2. Comparison of parameters between groups with and without hyperferritinemia.**

| Parameters | Overall subjects (n = 42,927) | | p value |
|---|---|---|---|
| | Normal ferritin (n = 34,743) | Hyperferritinemia (n = 8,184) | |
| Age (>39 years) | 15,723 (45.3) | 3,341 (40.8) | <0.001 |
| BMI (≥25 kg/m$^2$) | 11,880 (34.2) | 3,989 (48.7) | <0.001 |
| Non-smokers | 13,830 (39.8) | 3,122 (38.1) | 0.006 |
| No alcohol use | 4,390 (12.6) | 719 (8.8) | <0.001 |
| Hypercholesterolemia (≥220 mg/dL) | 6,055 (17.4) | 1,967 (24.0) | <0.001 |
| Hypertriglyceridemia (≥250 mg/dL) | 1,473 (4.2) | 826 (10.1) | <0.001 |
| High LDL cholesterol (≥159 mg/dL) | 4,458 (12.8) | 1,531 (18.7) | <0.001 |
| Elevated bilirubin (>1.9 mg/dL) (n = 42,926) | 578 (1.7) | 216 (2.6) | <0.001 |
| Elevated ALT (>40 U/L) (n = 42,925) | 940 (2.7) | 802 (9.8) | <0.001 |
| Elevated creatinine (>1.2 mg/dL) | 1,158 (3.3) | 309 (3.8) | 0.047 |
| Hyperglycemia at fasting (>100 mg/dL) | 8,674 (25.0) | 2,664 (32.6) | <0.001 |
| Elevated HbA1c (≥6.5%) (n = 42,921) | 211 (0.6) | 195 (2.4) | <0.001 |
| Elevated insulin (>25μIU/ml) (n = 42,871) | 66 (0.2) | 44 (0.5) | <0.001 |
| Elevated HOMA-IR (>1.31) (n = 42,871) | 16,522 (47.6) | 4,852 (59.3) | <0.001 |
| Elevated hsCRP (>0.5 mg/dL) (n = 34,979) | 522 (1.8) | 177 (2.7) | <0.001 |
| MBP (≥87 mmHg) (n = 42,924) | 14,578 (42.0) | 4,238 (51.8) | <0.001 |
| Quartile for FVC% predicted | | | 0.018 |
| Q1 (≤92%, n = 10,539, 24.6%) | 8,517 (24.5) | 2,022 (24.7) | |
| Q2 (93–97%, n = 9,203, 21.4%) | 7,350 (21.2) | 1,853 (22.6) | |
| Q3 (98–104%, n = 11,702, 27.3%) | 9,489 (27.3) | 2,213 (27.0) | |
| Q4 (≥105%, n = 11,483, 26.8%) | 9,387 (27.0) | 2,096 (25.6) | |
| Quartile for FEV1% predicted | | | <0.001 |
| Q1 (≤93%, n = 10,875, 25.3%) | 8,764(25.2) | 2,111 (25.8) | |
| Q2 (94–98%, n = 8,943, 20.8%) | 7,126 (20.5) | 1,817 (22.2) | |
| Q3 (99–105%, n = 11,627, 27.1%) | 9,426 (27.1) | 2,201 (26.9) | |
| Q4 (≥106%, n = 11,482, 26.8%) | 9,427 (27.1) | 2,055 (25.1) | |

Hyperferritinemia was defined as serum ferritin level > 300 ng/mL. Data are presented as number of subjects with percentage in parentheses. Continuous variables were transformed into categorical variables using median or mean cut-off values or upper normal limit values for the univariate analyses.

ALT = alanine aminotransferase; BMI = body mass index; FVC% predicted = percent predicted forced vital capacity; FEV1% predicted = percent predicted forced expiratory volume in 1s; HbA1c = hemoglobin A1c; HOMA-IR = homeostasis model assessment of insulin resistance; hsCRP = high-sensitivity C-reactive protein; LDL = low-density lipoprotein; MBP = mean blood pressure.

MBP = diastolic BP + (average systolic BP - average diastolic BP)/3.

quartiles of lung function and high iron in model 1 and model 2. Additionally, neither high iron nor TSAT was significantly associated with FEV1% or FVC% in model 3.

## Discussion

In this study, we demonstrated that hyperferritinemia is significantly associated with decreased FVC% and FEV1%, whereas iron and TSAT are not. To the best of our knowledge, this study is the first to describe an inverse association between lung function and hyperferritinemia.

Previously, four studies had evaluated the relationship between ferritin and lung function in the general population [9–12]. Contrary to our findings, a positive relationship between ferritin and lung function was found in two studies [9,10], although the other previous studies showed that lung function did not correlate with ferritin [11,12]. However, careful consideration is required when evaluating the relation between lung function and ferritin because

**Table 3. Univariate analyses to identify factors predictive of high iron and transferrin saturation.**

| Parameters | Iron (n = 35,890) | | | Transferrin saturation (n = 34,873) | | |
|---|---|---|---|---|---|---|
| | Low or normal iron (≤175 μg/dL) (n = 30,934) | High iron (>175 μg/dL) (n = 4,956) | p value | Low or normal TSAT (≤ 50%) (n = 24,378) | High TSAT (>50%) (n = 10,495) | p value |
| **Age (≥38 years)** | 14,328 (46.3) | 2,402 (48.5) | 0.005 | 11,095 (45.5) | 4,980 (47.5) | 0.001 |
| **BMI (>25 kg/m²)** | 11,534 (37.3) | 1,743 (35.2) | 0.004 | 9,400 (38.6) | 3,489 (33.2) | <0.001 |
| **Non-smoker** | 12,948 (41.9) | 1,512 (30.5) | <0.001 | 10,2296 (42.2) | 3,826 (36.5) | <0.001 |
| **Non-alcohol use** | 3,797 (12.3) | 425 (8.6) | <0.001 | 2,984 (12.2) | 1,094 (10.4) | <0.001 |
| **Hypercholesterolemia (≥220 mg/dL)** | 5,614 (18.1) | 903 (18.2) | 0.903 | 4,630 (19.0) | 1,652 (15.7) | <0.001 |
| **Hypertriglyceridemia (≥250 mg/dL)** | 1,637 (5.3) | 233 (4.7) | 0.082 | 1,419 (5.8) | 383 (3.6) | <0.001 |
| **High LDL cholesterol (≥159 mg/dL)** | 4,243 (13.7) | 614 (12.4) | 0.011 | 3,515 (14.4) | 1,147 (10.9) | <0.001 |
| **Elevated bilirubin (>1.9 mg/dL)** | 470 (1.5) | 215 (4.3) | <0.001 | 328 (1.3) | 341 (3.2) | <0.001 |
| **Elevated ALT (>40 U/L)** | 1,209 (3.9) | 253 (5.1) | <0.001 | 1,013 (4.2) | 397 (3.8) | 0.105 |
| **Elevated creatinine (>1.2 mg/dL)** | 1.011 (3.3) | 199 (4.0) | 0.007 | 794 (3.3) | 359 (3.4) | 0.433 |
| **Hyperglycemia at fasting (≥100 mg/dl)** | 7,814 (25.3) | 1,313 (26.5) | 0.064 | 6,309 (25.9) | 2,482 (23.6) | <0.001 |
| **Elevated HbA1c (≥6.5%)** | 286 (0.9) | 31 (0.6) | 0.037 | 243 (1.0) | 55 (0.5) | <0.001 |
| **Elevated insulin (>25 μIU/ml)** | 80 (0.3) | 7 (0.1) | 0.119 | 74 (0.3) | 1.0 (0.1) | <0.001 |
| **Elevated HOMA-IR (>1.31)** | 15,624 (50.5) | 2,250 (45.4) | <0.001 | 12,803 (52.6) | 4,557 (43.4) | <0.001 |
| **Elevated hsCRP (>0.5 mg/l)** | 688 (2.3) | 11 (0.2) | <0.001 | 633 (2.7) | 44 (0.4) | <0.001 |
| **MBP (≥86)** | 14,415 (46.6) | 2,488 (50.2) | <0.001 | 11,559 (47.4) | 4,798 (45.7) | 0.004 |
| **Quartile of FVC% predicted** | | | 0.097 | | | 0.011 |
| Q1 (≤92%) | 7,552 (24.4) | 1,167 (23.5) | | 5,965 (24.5) | 2,473 (23.6) | |
| Q2 (93–97%) | 6,648 (21.5) | 1,030 (20.8) | | 5,292 (21.7) | 2,187 (20.8) | |
| Q3 (98–104%) | 8,376 (27.1) | 1,399 (28.2) | | 6,556 (26.9) | 2,939 (28.0) | |
| Q4 (≥105%) | 8,358 (27.0) | 1,360 (27.4) | | 6,565 (26.9) | 2,896 (27.6) | |
| **Quartile of FEV1% predicted** | | | 0.387 | | | 0.061 |
| Q1 (≤93%) | 7,801 (25.2) | 1,243 (25.1) | | 6,152 (25.2) | 2,594 (24.7) | |
| Q2 (92–98%) | 6,384 (20.6) | 1,019 (20.6) | | 5,088 (20.9) | 2,120 (20.0) | |
| Q3 (99–105%) | 8,436 (27.3) | 1,305 (26.3) | | 6,585 (27.0) | 2,879 (27.4) | |
| Q4 (≥106%) | 8,313 (26.9) | 1,389 (28.0) | | 6,553 (26.9) | 2,902 (27.7) | |

High iron and high transferrin saturation were defined as an iron level > 175 μg/dL and transferrin saturation > 50%, respectively. Data are presented as the number of subjects with percentage in parenthesis. Continuous variables were transformed into categorical variables using median or mean cut-off values or upper normal limit values for the univariate analyses.

ALT = alanine aminotransferase; BMI = body mass index; FVC% predicted = percent predicted forced vital capacity; FEV1% predicted = percent predicted forced expiratory volume in 1s; HbA1c = hemoglobin A1c; HOMA-IR = homeostasis model assessment of insulin resistance; hsCRP = high-sensitivity C-reactive protein; LDL = low-density lipoprotein; MBP = mean blood pressure; TSAT = transferrin saturation.

MBP = diastolic BP + (average systolic BP - average diastolic BP)/3.

various parameters, including inflammation markers and cardio-metabolic diseases, are associated with both ferritin level [3,10,13,14,16,24,34–36] and lung function [4,5,37,38]. Considering that, our study clearly differs from the previous studies [9–12]. The serum ferritin levels in this study were higher (199.3 ng/mL) than in the previous studies (36.4–129.3 ng/mL), possibly because of differences in age and sex distributions of the study participants. The median age of the enrolled subjects in this study was younger than in previous studies [9,10,12]. Furthermore,

**Table 4. Multiple logistic regression analysis of the odds of hyperferritinemia, high transferrin saturation, and high iron by quartile of lung function.**

| | Model 1 | | | Model 2 | | | Model 3 | | |
|---|---|---|---|---|---|---|---|---|---|
| | OR (95% CI) | p value | p for trend | OR (95% CI) | p value | p for trend | OR (95% CI) | p value | p for trend |
| **Hyperferritinemia (n = 42,927)** | | | | | | | | | |
| **Quartile of FVC% predicted** | | | <0.001 | | | <0.001 | | | 0.001 |
| **Q1 (≤92%, n = 10,539, 24.6%)** | 1.145 (1.062–1.234) | <0.001 | | 1.157 (1.073–1.247) | <0.001 | | 1.150 (1.056–1.252) | 0.001 | |
| **Q2 (93–97%, n = 9,203, 21.4%)** | 1.137 (1.064–1.216) | <0.001 | | 1.146 (1.072–1.226) | <0.001 | | 1.101 (1.021–1.188) | 0.013 | |
| **Q3 (98–104%, n = 11,702, 27.3%)** | 1.087 (1.019–1.159) | 0.011 | | 1.090 (1.023–1.163) | 0.008 | | 1.094 (1.018–1.176) | 0.015 | |
| **Q4 (≥105%, n = 11,483, 26.8%)** | 1 | | | 1 | | | 1 | | |
| **Quartile of FEV1% predicted** | | | <0.001 | | | <0.001 | | | 0.007 |
| **Q1 (≤93%, n = 10,875, 25.3%)** | 1.164 (1.085–1.248) | <0.001 | | 1.167 (1.088–1.251) | <0.001 | | 1.140 (1.053–1.233) | 0.001 | |
| **Q2 (94–98%, n = 8,943, 20.8%)** | 1.133 (1.049–1.223) | 0.001 | | 1.138 (1.054–1.229) | 0.001 | | 1.100 (1.007–1.200) | 0.033 | |
| **Q3 (99–105%, n = 11,627, 27.1%)** | 1.098 (1.030–1.171) | 0.004 | | 1.098 (1.030–1.171) | 0.004 | | 1.081 (1.005–1.163) | 0.035 | |
| **Q4 (≥106%, n = 11,482, 26.8%)** | 1 | | | 1 | | | 1 | | |
| **High iron (n = 37,873)** | | | | | | | | | |
| **FVC% predicted** | | | 0.087 | | | 0.396 | | | 0.901 |
| **Q1: ≤92% (n = 8,719, 24.3%)** | 0.947 (0.871–1.031) | 0.209 | | 0.981 (0.901–1.067) | 0.652 | | 1.011 (0.927–1.103) | 0.802 | |
| **Q2: 93–97% (n = 7,678, 21.4%)** | 0.950 (0.870–1.036) | 0.247 | | 0.974 (0.892–1.063) | 0.549 | | 0.995 (0.910–1.089) | 0.916 | |
| **Q3: 98–104% (n = 9,775, 27.2%)** | 1.024 (0.945–1.110) | 0.566 | | 1.036 (0.956–1.123) | 0.391 | | 1.052 (0.969–1.143) | 0.229 | |
| **Q4: ≥105% (n = 9,718, 27.1%)** | 1 | | | 1 | | | 1 | | |
| **FEV1% predicted** | | | 0.366 | | | 0.477 | | | 0.980 |
| **Q1: ≤93% (n = 9,044, 25.2%)** | 0.925 (0.853–1.004) | 0.062 | | 0.928 (0.855–1.007) | 0.074 | | 0.947 (0.871–1.030) | 0.203 | |
| **Q2: 92–98% (n = 7,403, 20.6%)** | 0.953 (0.873–1.040) | 0.277 | | 0.960 (0.880–1.048) | 0.363 | | 0.994 (0.908–1.087) | 0.887 | |
| **Q3: 99–105% (n = 9,741, 27.1%)** | 0.953 (0.877–1.035) | 0.249 | | 0.959 (0.883–1.042) | 0.327 | | 0.987 (0.907–1.075) | 0.765 | |
| **Q4: ≥106% (n = 9,702, 27.0%)** | 1 | | | 1 | | | 1 | | |
| **High TSAT (n = 38,818)** | | | | | | | | | |
| **FVC% predicted** | | | 0.002 | | | 0.014 | | | 0.227 |
| **Q1: ≤92% (n = 8,438, 24.2%)** | 1.011 (0.950–1.075) | 0.732 | | 1.017 (0.956–1.082) | 0.589 | | 1.033 (0.970–1.101) | 0.310 | |
| **Q2: 93–97% (n = 7,479, 21.4%)** | 0.930 (0.870–0.994) | 0.032 | | 0.942 (0.882–1.007) | 0.080 | | 0.975 (0.911–1.044) | 0.467 | |
| **Q3: 98–104% (n = 9,495, 27.2%)** | 0.924 (0.866–0.985) | 0.016 | | 0.941 (0.882–1.003) | 0.063 | | 0.975 (0.913–1.042) | 0.456 | |
| **Q4: ≥105% (n = 9,461, 27.1%)** | 1 | | | 1 | | | 1 | | |
| **FEV1% predicted** | | | 0.062 | | | 0.083 | | | 0.772 |
| **Q1: ≤93% (n = 8,746, 25.1%)** | 0.986 (0.927–1.049) | 0.659 | | 0.988 (0.929–1.051) | 0.707 | | 1.006 (0.944–1.072) | 0.857 | |
| **Q2: 92–98% (n = 7,208, 20.7%)** | 0.942 (0.881–1.007) | 0.079 | | 0.945 (0.884–1.011) | 0.101 | | 0.982 (0.917–1.052) | 0.604 | |
| **Q3: 99–105% (n = 9,464, 27.1%)** | 0.952 (0.893–1.014) | 0.125 | | 0.955 (0.896–1.018) | 0.158 | | 0.997 (0.934–1.064) | 0.930 | |

*(Continued)*

**Table 4.** (Continued)

| | Model 1 | | | Model 2 | | | Model 3 | | |
|---|---|---|---|---|---|---|---|---|---|
| | OR (95% CI) | p value | p for trend | OR (95% CI) | p value | p for trend | OR (95% CI) | p value | p for trend |
| Q4: ≥106% (n = 9,455, 27.1%) | 1 | | | 1 | | | 1 | | |

Model 1 was adjusted for age, BMI, and MBP. Model 2 was adjusted as in model 1 plus smoking and alcohol. Model 3 was adjusted as in model 2 plus variables with a *P* value < 0.05 in the univariate analyses (liver function test, lipid battery, hsCRP level, glucose level, insulin level, HbA1c level, and HOMA-IR).

CI = confidence interval; FVC% predicted = percent predicted forced vital capacity; FEV1% predicted = percent predicted forced expiratory volume in 1s;

HbA1c = hemoglobin A1c; HOMA-IR = homeostasis model assessment of insulin resistance; hsCRP = high-sensitivity C-reactive protein; OR = odds ratio;

TSAT = transferrin saturation.

all the previous studies enrolled females, who represented more than half of the total number of subjects, in contrast to this study. Although serum ferritin decreases with age in males and increases with age in females, the factor with the greatest effect on the discrepant level of ferritin is sex [24,36]. Hormonal effects and increased iron loss can cause physiological differences in iron homeostasis and biomarker distributions between males and females. Consequently, sex segregation analyses are needed to yield meaningful results and minimize potential confounding factors. Moreover, the previous studies included subjects with cardio-metabolic diseases [9,12]. In Brigham's study [11], about 16% of the subjects had a history of asthma, while subjects with FEV1/FVC < 0.7 were included in Lee et al.'s study (13.4%). [10] Because FEV1/FVC < 0.7 predominantly reflects obstruction of middle sized airways, it is possible that Lee et al.'s study [10] contained more subjects with functional and structural lung debilities than ours. Ghio's study [9] evaluated the correlation between lung function and ferritin without adjusting for confounding factors, and two studies [10,11] adjusted for only a few variables, without adjusting for other relevant confounders that affect ferritin level [3,13,14,16] and lung function [4,5,37–39]. Failing to adjust for those confounders could have distorted the outcomes of the previous studies. We excluded subjects with overt cardio-metabolic and pulmonary diseases. Additionally, we adjusted for many relevant confounders associated with ferritin and lung function and examined only males. After fully adjusting for potential confounders, we found a robust negative association between hyperferritinemia and lung function in healthy men. These results indicate that healthy subjects with hyperferritinemia may have early perturbations of lung function. This finding is important. Reduced lung function is a marker of an individual's increased susceptibility to COPD and a major risk factor for cardiovascular morbidity and mortality, which are potentially preventable diseases with significant health and economic impacts worldwide [40]. Further, reduced lung function is a powerful predictor of mortality [41]. Therefore, the current study is important, given the projected growing public health impact of lung function and outdoor air pollution [6].

The mixed results from the previous studies [9,10] and this study complicate a conclusion on whether ferritin may have a beneficial or noxious effect on lung function. The reasons for the mixed results are unclear. However, serum ferritin level in a different composition of study subjects might explain the disagreement. Actually, serum ferritin level in our study was about three times higher than those in previous studies. Interestingly, previous studies showed a threshold effect of ferritin with incident type 2 diabetes [42,43]. Like this, the biological effects of ferritin on lung function may also be attributed to a threshold effect in which the cohort including more individuals with low serum ferritin level showed a positive association [9,10] between lung function and ferritin level by enhancing detoxification of iron, and vice versa. There is a biologically plausible explanation for this relationship and our findings. Iron must bind to proteins to prevent tissue damage from free radical formation [1,3]. Ferritin is a key

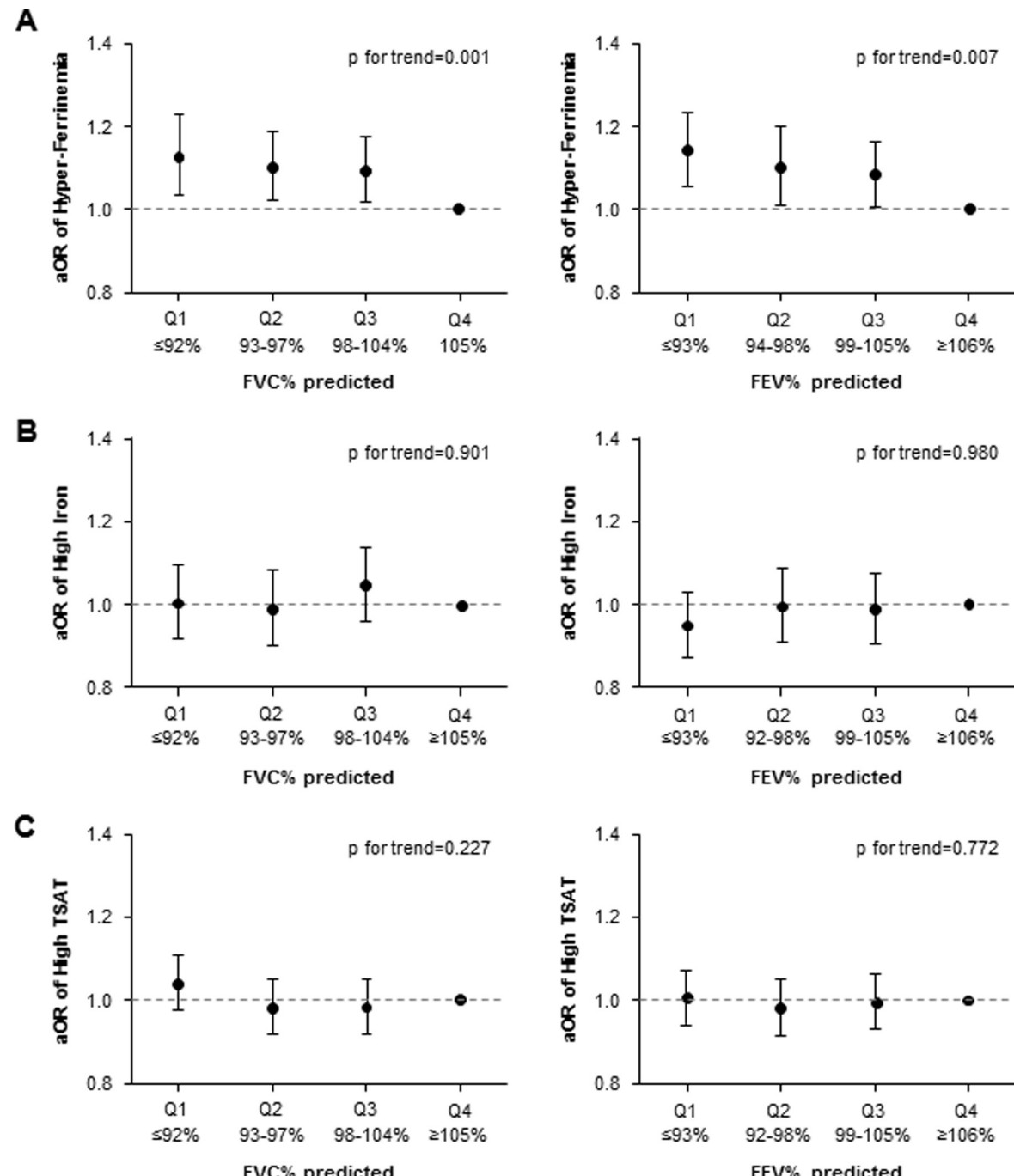

**Fig 2. Multivariable-adjusted odds ratio (aOR) for high biomarkers of iron metabolism according to quartile of lung function.** The aOR for hyperferritinemia increased with decreasing quartiles of FEV1% and FVC% (A) in a dose-response manner. The reference values were set as the highest quartile of FEV1% and FVC%. Neither high iron nor TSAT was significantly associated with FEV1% or FVC% (B and C). Models were adjusted for potential covariates and metabolic laboratory markers (age, BMI, mean blood pressure, alcohol intake, smoking, liver function test, lipid battery, hsCRP level, glucose level, insulin level, HbA1c level, and HOMA-IR). FVC% predicted = percent predicted forced vital capacity; FEV1% predicted = percent predicted forced expiratory volume in 1s; HbA1c = hemoglobin A1c; HOMA-IR = homeostasis model assessment of insulin resistance; hsCRP = high-sensitivity C-reactive protein; OR = odds ratio; TSAT = transferrin saturation.

protein in iron homeostasis and presents a paradox. The capacity of ferritin to prevent iron's pro-oxidant activity by oxidizing and sequestering the metal suggests that it might play an important role as an antioxidant within the range of homeostasis and be a marker of iron-related oxidative toxicity after disruption of iron homeostasis [1,3,8]. Consequently, a delicate homeostasis of iron is vitally important, perhaps more so in lung than in any other organs, given the compounded damage of ferritin and high local oxygen tensions in the lung. If exceeding the limit for cellular iron regulation irrespective of iron store regulation, ferritin released from damaged cells could result in elevated serum ferritin concentration, losing most of its iron and leaving free iron [3]. Free iron beyond what the body can adequately detoxify could accumulate both locally in the airway epithelium and systemically [7], causing subsequent oxidative damage [1,3], through superoxide generation and leading to permanent loss of lung function over time [4,44]. Thus, iron-related oxidative damage might be the link between hyperferritinemia and early perturbations in lung function. The biological effects of ferritin on lung function may be attributed to a threshold effect on protection from respiratory damage.

On the other hand, previous studies showed that serum iron was positively associated with lung function [9,11,12]. Brigham's study [11] showed that higher ferritin was associated with a lower risk of asthma only in the reference range strongly correlated with iron storage (20–300 ng/ml) [45]. The question remains whether elevated serum iron has a protective or detrimental effect on the lungs. Lung inflammation in response to noxious inhalants could involve active repletion from serum iron [7]. Continually decreased iron seems to worsen hypoxemia, producing lung functional and structural debilities [46]. However, iron is potentially hazardous in the range that exceeds the iron-detoxification capacity of ferritin [1]. Therefore, tight regulation over iron metabolism is necessary to prevent both iron deficits and overloads. Perhaps only subjects with homeostatic ferritin level that counteracts free iron are protected against the decline in lung function caused by continuous exposure to oxidative stress. However, we observed a threshold effect above levels within the reference range for biomarkers of iron metabolism on the lung function. Unlike in previous studies [9,11,12], we found that high iron and TSAT were not significantly associated with lung function in the fully adjusted models that included inflammatory markers and cardio-metabolic risk factors. Similarly, several studies adjusting for various confounders showed that ferritin was an independent risk factor for cardio-metabolic diseases [13–18], whereas other biomarkers of iron metabolism were not always statistically significant, especially in men [13–17]. This reflects the possibility that ferritin could affect lung function and cardio-metabolic diseases regardless of a body's iron status. Ferritin is not only a marker of body iron stores, but also an acute-phase reactant that can fluctuate in response to inflammatory mediators and metabolic stress [3]. Systemic inflammation could be an important mediator of hyperferritinemia [42,47] and decreased lung function [4]. However, the association of lung function with hyperferritinemia remained significant in the current study, even after adjusting for several other markers related to systemic inflammation, in line with previous studies [14,42]. The bioavailable iron is responsible for reactive oxygen species (oxidative stress) and IL-8 induction (inflammatory mediator) [8]. IL-8 is highly chemo-attractive for neutrophils, eosinophils, and T lymphocytes [48,49], in contrast to IL-6, which is the chief stimulator of CRP production. This might indicate that ferritin increases the risk of decline in lung function through metabolic oxidative stress other than systemic inflammation. Therefore, ferritin is a robust biomarker for lung function in our study population, but no associations were noted for other biomarkers of iron metabolism with lung function. These results lead us to postulate that the elevated ferritin levels seen with cardio-metabolic diseases and decreased lung function could result primarily from metabolic stress that reflects oxidative damage independent of iron stores.

Our study has several strengths and limitations. A major strength of our study is its large sample size, with subjects drawn from a healthy population without overt disease. Another strength is that we conducted analyses adjusting for various confounders that affect lung function and ferritin. This gave us sufficient statistical power and could be more relevant to healthy population with normal lung functions. However, it should be considered that it is possible to show statistical significance for small differences in the large sample sized studies. And, our study demonstrated modest association between hyperferritinemia and lung function parameters. There also have been discordant results on this issue among studies [9–12], although those were undertaken in populations where the serum ferritin level were lower than the current study. It is suggested that a variety of factors are likely to have more influence on lung function than ferritin in the real world, and vice versa. Therefore, we cannot exclude the possibility of some unmeasured or residual confounding factors in the association between ferritin and lung function. Furthermore, there has been no consensus on criteria for hyperferritinemia and, there could be debate about whether our definition of hyperferritinemia may be appropriate threshold for protection from respiratory damage. Even if our study with the large sample size demonstrated significant association between hyperferritinemia and lung function parameters, further prospective studies are needed to determine whether ferritin is an independent risk of lung function deterioration and what the threshold level is appropriate. Our study also had several limitations. First, we cannot infer causation due to the cross-sectional nature of our study. Therefore, further studies are needed to elucidate the precise mechanism underlying the phenomena we observed. Second, there is the possibility of selection bias in participant recruitment because the study participants were mostly middle-aged Korean men in an urban community who all received a health screening at a single university hospital. Therefore, our findings cannot be generalized to other populations or ethnic groups. Third, we measured biomarkers of iron metabolism, including ferritin, at a single time point. Therefore, we cannot exclude the possibility of intra-individual changes in those levels because ferritin varies widely within an individual's lifetime [36], and iron shows diurnal variation without changes in total body iron [50]. Fourth, the true incidence of pulmonary and cardio-metabolic diseases could have been underestimated in our study due to the questionnaire-based collection of medical histories. Some individuals with subclinical disease processes could have also be included, although we excluded subjects with overt medical diseases and ventilation pattern. Such inclusion might be significant, because the relationship between subclinical pulmonary and cardio-metabolic disease can contribute to a decline in lung function, especially among individuals with abnormal ferritin level. Finally, this study was not hospital-based. Lack of data on genetic and environmental variations that could have influenced our results, including daily dietary iron, is another potential limitation of this study. Additionally, lack of measurements of oxygen saturation and partial pressure of oxygen in the blood (PaO$_2$), which could affect iron metabolism [51], are also potential limitations, although we excluded those with any known evidence of clinical medical diseases including lung disease.

In conclusion, hyperferritinemia was independently associated with decreased lung function because of inflammatory mechanisms independent of iron stores. However, longitudinal follow-up studies and prospective interventional studies are needed to validate our findings.

## Author Contributions

**Conceptualization:** Jae-Uk Song.

**Data curation:** Jonghoo Lee, Hye kyeong Park, Jae-Uk Song.

**Formal analysis:** Jonghoo Lee, Hye kyeong Park, Jae-Uk Song.

**Investigation:** Jae-Uk Song.

**Methodology:** Min-Jung Kwon, Joon Mo Kim.

**Supervision:** Jae-Uk Song.

**Writing – original draft:** Jonghoo Lee, Hye kyeong Park, Jae-Uk Song.

**Writing – review & editing:** Min-Jung Kwon, Soo-Youn Ham, Joon Mo Kim, Si-Young Lim, Jae-Uk Song.

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
