## [Decision Letter · Decision Letter 0]

11 Nov 2019

PONE-D-19-26396

Decreased lung function is associated with elevated ferritin, but not iron or transferrin saturation in 47,981 healthy Korean men: a cross-sectional study

PLOS ONE

Dear Dr. Song,

Thank you for submitting your manuscript to PLOS ONE. After careful consideration, we feel that it has merit but does not fully meet PLOS ONE’s publication criteria as it currently stands. Therefore, we invite you to submit a revised version of the manuscript that addresses the points raised during the review process.

The manuscript has been assessed by two reviewers; their comments are available below.

One of the reviewers is positive about the study but the second has raised some methodological concerns. The reviewer notes concerns about the exclusion of participants with any lung condition and about the exclusion of women from the sample, the reviewer recommend further analyses including the women population. The reviewer also notes concerns about the FVC and FEV1 results reported, and both reviewers recommend more in-depth discussion of related literature.

Could you please carefully revise the manuscript to address the concerns raised by the reviewers?

We would appreciate receiving your revised manuscript by Dec 24 2019 11:59PM. Please include the following items when submitting your revised manuscript:

We look forward to receiving your revised manuscript.

Kind regards,

Iratxe Puebla

Senior Managing Editor, PLOS ONE

Journal Requirements:

1. We noticed you have some minor occurrence(s) of overlapping text with the following previous publication(s), which needs to be addressed:

https://doi.org/10.1371/journal.pone.0208736

https://doi.org/10.1111/resp.13370

https://doi.org/10.1080/14397595.2017.1285981

In your revision ensure you cite all your sources (including your own works), and quote or rephrase any duplicated text outside the Methods section. Further consideration is dependent on these concerns being addressed.

Reviewers' comments:

Reviewer's Responses to Questions

**Comments to the Author**

1. Is the manuscript technically sound, and do the data support the conclusions?

Reviewer #1: Yes

Reviewer #2: Partly

2. Has the statistical analysis been performed appropriately and rigorously? 

Reviewer #1: Yes

Reviewer #2: Yes

3. Have the authors made all data underlying the findings in their manuscript fully available?

Reviewer #1: Yes

Reviewer #2: No

4. Is the manuscript presented in an intelligible fashion and written in standard English?

Reviewer #1: Yes

Reviewer #2: Yes

5. Review Comments to the Author

Reviewer #1: Authors in this study conducted a cohort study of 47,981 healthy Korean men, and found hyperferritinemia was associated with decreased lung function in healthy Korean men, whereas iron and TSAT were not. In general, this finding is interesting and useful, even though longitudinal follow-up studies and prospective interventional studies are needed to validate it. It better to discuss the relationship between iron and oxygen concentration in discussion, as it was mentioned previously that different iron need under different oxygen concentration. (Cancer Lett. 2019 Nov 1;464:56-61.)

Reviewer #2: Overview

This study looks at the relationship of ferritin, iron and iron saturaturation in relationship to lung function and other parameters of systemic inflammation. The study population is ~47000 men representing ~25% of a large screening program from a single Korean hospital, after excluding all with history of any disease or taking any medication. Unlike other, population based, studies of this question which have looked at correlations to the spectrum of ferritin levels, the authors have chosen to focus on the likelihood of a ferritin level above the upper limit of normal (ULN). Women were excluded because <1% exceeded the chosen ULN. The results in relationship to lung function appear discordant with prior studies and potential reasons for this are discussed.

Major comments:

1.The study premise is that elevated ferritin is associated with systemic inflammation and pulmonary injury and thus may be associated with decreased lung function, in what would presumably be a pathologic process. However, the study population excluded those with any known evidence of lung disease so the lung function data available are nearly all within the normal range of a healthy population where most of the variation above and below the predicted value is considered to be random rather than pathologic. Certainly, some individuals with early disease processes will have low normal values, but this choice would seem to create a low signal to noise situation. The rationale for the choice to exclude even mild airflow limitation from the study population should be explained.

2.Since women are known to have different ferritin levels and behaviors (eg, with age) than men, it is unfortunate that this population was excluded by the choice to consider only hyper-ferritinemia rather than a median, mean or correlation to the range of values. Since this was essentially a normal, disease free population one might expect 2.5% to be above the ULN (assuming a 95% range and false positives at both ends) so the finding of <1% suggest the ULN chosen for women may be too high. Similarly, the finding of nearly 20% of these healthy men to be above the ULN (rather than the expected 2.5%) needs some discussion, starting with the source and derivation of these cut points (general population or “healthy” pop, smokers excluded or not, appropriate to Korean ethnicity?). An alternative analysis of “hyper” ferritin, that would allow inclusion of women, would be to divide the groups at the 80th %iles for men and women of the study population.

3.The FVC and FEV1 results are transposed between the results stated in the Abstract and those shown in Table 1. Whichever is correct, it would be very unusual in a near normal population to have a 20% difference (93% vs 113%) between the mean %predicted FEV1 and FVC. This would mean that the average FEV1/FVC ratio would have to be either very high or very low, but the reported value of 82 is normal, assuming that is the actual ratio (not %pred). Preferably, it could be reported as 0.82 to eliminate any potential uncertainty for the reader. It seems likely that there is an error in the formula or calculation of the predicted values (although it is recognized that, if the error is consistent, this would not affect the quartile analysis). It would be helpful to add the actual mean FVC and FEV1 values to Table 1, and the authors also might consider comparing these predictions to the North Asia data from the GLI-2012 predictions (Quanjer. Eur Respir J 2012; 40:1324-43).

(Although I cannot access Ms ref 17, the prediction equations reported in the Ms appear to be correct. Calculations for a 40yo, 170cm, 70kg male result in reasonable values for FVC 4.78, FEV1 3.97 for a ratio of 0.83, so for this example an FVC of 100%pred and an FEV1 of 100%pred would give a ratio of .83, but 93%-113%, either way, would be very different).

4.DISCUSSION. The discussion of the different findings between this study and Ms refs 5 and 6 should note that these population-based studies looked at lung function in relationship to the full range of ferritin values or to mean/median values, whereas this study looks at the likelihood of very high values. While it is reasonable to think that these two approaches might correlate, that is uncertain and should be discussed along with stating the rationale for the choice made for this study. This data could also be analysed in a fashion similar to ref 5 to make a more ‘apples to apples’ comparison.

5.CONCLUSION. The introduction proposes a dichotomy of hyper-ferritinemia as a marker of iron stores vs as a marker of systemic inflammation. The absence of an association of iron and iron saturation with lower lung function leads to the conclusion (p24 line 150) that the observed relationship of hyper-ferritinemia to lower lung function is “a result of inflammation”. However, Model 3 adjusted for several other markers of systemic inflammation which might be expected to eliminate the association, but instead only reduced the positive odds ratios modestly (21 and 30% for FEV1 and FVC). The implications of this deserves comment in Discussion.

Minor Comments:

Table 1 Ferritin level by quartiles for FEV1 shows units of ng/mL while for FVC and elsewhere mg/dL is used. Terminology should be consistent throughout the Ms, tables and figures.

Table 2 It appears that no exclusions were made based upon lung function data. <3% with FEV1/FVC <70 were included. No mention is made of possible early restrictive disease (which has also been associated with higher ferritin levels) in the study population. With a mean FVC of 92+/-10 %pred and a 1st quartile < 86%pred it is likely that some would meet a spirometric criteria for restriction, but as noted in comment 3 above the accuracy of those %pred values is uncertain.

p21 line 67 The sentence “Although a positive relationship between ferritin and lung function was found…” is accurate, but may not convey its meaning to the reader – could it mean a positive relationship with a decrement in lung function? (I pulled the ref to be sure.) Perhaps the sentence might begin “Contrary to our findings, a positive relationship…” or end explicitly “….two studies, with increased ferritin associated with higher lung function.”

p22 line 95 Since this study is not designed to show causality, it would be more accurate to state “whether ferritin may have a beneficial or noxious effect…”

6. PLOS authors have the option to publish the peer review history of their article (what does this mean?). If published, this will include your full peer review and any attached files.

Reviewer #1: No

Reviewer #2: No

---

## [Author Response · Author response to Decision Letter 0]

21 Dec 2019

We already uploaded the file named "Response to Reviewers" 

Actually our response is too long to respond in this area 

Wolud you please refer to reference (as we attach PDF files) for your comment 3

---

## [Decision Letter · Decision Letter 1]

6 Feb 2020

PONE-D-19-26396R1

Manuscript ID: PONE-D-19-26396

Decreased lung function is associated with elevated ferritin, but not iron or transferrin saturation in 42,927 healthy Korean men: a cross-sectional study

PLOS ONE

Dear Dr. Song,

Thank you for submitting your manuscript to PLOS ONE. After careful consideration, we feel that it has merit but does not fully meet PLOS ONE’s publication criteria as it currently stands. Therefore, we invite you to submit a revised version of the manuscript that addresses the points raised during the review process.

Specifically, the reviewer 2 still had some concerns that need to be adequately addressed. I hope the authors can effectively respond to these comments.

We would appreciate receiving your revised manuscript by Mar 22 2020 11:59PM. To enhance the reproducibility of your results, we recommend that if applicable you deposit your laboratory protocols in protocols.io, where a protocol can be assigned its own identifier (DOI) such that it can be cited independently in the future. For instructions see: http://journals.plos.org/plosone/s/submission-guidelines#loc-laboratory-protocols

We look forward to receiving your revised manuscript.

Kind regards,

Yu Ru Kou, PhD

Academic Editor

PLOS ONE

Reviewers' comments:

Reviewer's Responses to Questions

**Comments to the Author**

1. If the authors have adequately addressed your comments raised in a previous round of review and you feel that this manuscript is now acceptable for publication, you may indicate that here to bypass the “Comments to the Author” section, enter your conflict of interest statement in the “Confidential to Editor” section, and submit your "Accept" recommendation.

Reviewer #1: All comments have been addressed

Reviewer #2: (No Response)

2. Is the manuscript technically sound, and do the data support the conclusions?

Reviewer #1: Yes

Reviewer #2: Yes

3. Has the statistical analysis been performed appropriately and rigorously? 

Reviewer #1: Yes

Reviewer #2: I Don't Know

4. Have the authors made all data underlying the findings in their manuscript fully available?

Reviewer #1: Yes

Reviewer #2: No

5. Is the manuscript presented in an intelligible fashion and written in standard English?

Reviewer #1: Yes

Reviewer #2: Yes

6. Review Comments to the Author

Reviewer #1: Authors in this study conducted a cohort study of 47,981 healthy Korean men, and found hyperferritinemia was associated with decreased lung function in healthy Korean men, whereas iron and TSAT were not.

The authors have revised this manuscript carefully,and there are no question now.

Reviewer #2: The revised study has been modified by the exclusion of subjects with low measured lung function, just over 10% of the prior study group. A new source of reference equations has been chosen resulting in %predicted values which are much more reasonable than the prior report.

Specific Comments

P4 Study Design The criteria used for PFT exclusion should be stated

(eg, FVC <80%, FEV1<80%, FEV1/FVC <0.70).

P5 Lung Fxn measurement Thank you for changing the Reference data source (although I think there was also a calculation error in the prior version); the predicted values now look much more reasonable. The source is cited (and available online) so it is not necessary to include the equations in the Ms.

The last sentence re FEV1/FVC can be deleted here and stated earlier (prior comment).

Results With the large number of subjects it is possible to show statistical significance for very small differences. Whether or not these are clinically or physiologically important is another matter that should be addressed. This would be most appropriate in Discussion, but the presentation of results can also be tempered.

eg, “subjects in the hyperferritinemia group were more likely to be younger”, yet a mean age of 38.7 v 38.1, with wide overlap of SDs, is hardly compelling. Similarly, for smoking (~62% v 60%) and alcohol use (91.2 v 87.4%). The message might be how similar these exposures are despite the wide difference in ferritin levels. These results would be appropriately preceded by “Small but significant differences were seen….” or similar acknowledgement of the small magnitude.

More importantly, the differences in the primary comparison of lung function are also very small in both absolute terms (FVC 4.75 v 4.73; FEV1 3.89 v 3.88) and when adjusted for age and height by % predicted. The latter should be reported to at least one additional place so that rounding does not influence the apparent difference. It seems mathematically odd here that the FVC and FEV1 values shown for the combined groups match that of the smaller hyper- group rather than that of the normal group making up over 80% of the study population.

Rounding is also an issue for bilirubin and creatinine, where the reported numbers for the two groups appear identical, but for bilirubin the star indicates a highly significant difference.

The lower portion of Table 1 shows a slight inverse relationship of the ferritin levels vs FVC and FEV1 quartiles. This is seen in both the normal and hyper ferritin groups, contrary to the idea that there may be a threshold effect for ferritin to affect lung function. It is not clear what the significance test (“compared with hyperferritinemia”) means here; is there a difference in the relationship between the two?...and if so, which is stronger? For both FVC and FEV1, the % difference in ferritin level from quartile 1 to 4 is greater for the normal group, although the absolute differences are larger in the hyper- group.

Table 2

The % numbers in title are confusing and should be removed. They show the distribution between groups, while the similarly displayed % values in the table do not, but are based upon the total in each group.

7. PLOS authors have the option to publish the peer review history of their article (what does this mean?). If published, this will include your full peer review and any attached files.

Reviewer #1: No

Reviewer #2: No

---

## [Author Response · Author response to Decision Letter 1]

21 Feb 2020

TO THE COMMENTS OF THE REVIEWER 2.

The revised study has been modified by the exclusion of subjects with low measured lung function, just over 10% of the prior study group. A new source of reference equations has been chosen resulting in %predicted values which are much more reasonable than the prior report.

Specific comments:

C1. Study Design 

The criteria used for PFT exclusion should be stated (eg, FVC <80%, FEV1<80%, FEV1/FVC< 0.70).

The last sentence re FEV1/FVC can be deleted here and stated earlier (prior comment).

R1.

We do appreciate the reviewer’s comment. We apologize for our carelessness and the lack of clarity.

As reviewer pointed out, we stated the criteria used for PFT exclusion in the manuscript as follows; 

(from line 85 to line 86 on page 4 );

We excluded 85,455 participants who had a ventilation disorder on the basis of spirometric results [1]



We excluded 85,455 participants who had an impaired lung function (a ventilation disorder) (the subjects without normal lung function defined as forced expiratory volume in one second (FEV1) to forced vital capacity (FVC) [FEV1/FVC] ≥ 0.7 and FVC ≥80% predicted) [2] 

C2. Lung Fxn measurement 

Thank you for changing the Reference data source (although I think there was also a calculation error in the prior version); the predicted values now look much more reasonable. The source is cited (and available online) so it is not necessary to include the equations in the Ms. The last sentence re FEV1/FVC can be deleted here and stated earlier (prior comment).

R2.

Thank you for your advice to improved our manuscript much more. As you recommended, we removed the reference equations in the manuscript. (from line 138 to line 139 on page 6)

And, we apologize for our carelessness. Thank you for pointing our mistakes.

We omitted last sentence in the lung function measurement section. (from line 143 on page 6) 

C3. Results 

C3-1.

With the large number of subjects it is possible to show statistical significance for very small differences. Whether or not these are clinically or physiologically important is another matter that should be addressed. This would be most appropriate in Discussion, but the presentation of results can also be tempered.

eg, “subjects in the hyperferritinemia group were more likely to be younger”, yet a mean age of 38.7 v 38.1, with wide overlap of SDs, is hardly compelling. Similarly, for smoking (~62% v 60%) and alcohol use (91.2 v 87.4%). The message might be how similar these exposures are despite the wide difference in ferritin levels. These results would be appropriately preceded by “Small but significant differences were seen….” or similar acknowledgement of the small magnitude.

More importantly, the differences in the primary comparison of lung function are also very small in both absolute terms (FVC 4.75 v 4.73; FEV1 3.89 v 3.88) and when adjusted for age and height by % predicted. The latter should be reported to at least one additional place so that rounding does not influence the apparent difference. 

R3-1.

Thank you for your reasonable comments. The reviewer makes a very important point. We fully understand reviewer’s concern. We also agree that it could not be clinically meaningful, although it show statistical significance in a large study population. So, we need to tone the presentation of results down, as reviewer pointed out. We modified result section to reflect reviewer’s concern.

(from line 168 to line 172 on page 7 );

Compared with the normal ferritin group, subjects in the hyperferritinemia group were more likely to be younger, to have smoked, to drink alcohol, and have higher blood pressure. Serum ferritin level was negatively associated with a quartile increase in FVC% (p=0.001) and FEV1% (p<0.001), but the difference in FEV1 (L)/FVC (L) between the groups was not statistically significant (p=0.797). 



When compared clinical variables between two groups, small but significant differences were seen in age, smoking habit, alcohol intake, liver function, CRP, blood pressure and a variety of metabolic parameters, including BMI, fasting glucose, and HbA1c. Compared with the normal ferritin group, subjects in the hyperferritinemia group were to have lower value of spirometry with a narrow margin, although those were statistically significant. However, the difference in FEV1 (L)/FVC (L) between the groups was not statistically significant (p=0.797).

(from line 176 on page 7 to line 180 on page 8 );

A comparison of clinical characteristics between subjects with and without high iron or TSAT is shown in Table 3. Subjects with high iron and TSAT were more likely than others to drink and smoke. However, both high iron and TSAT were inversely associated with hsCRP and metabolic values, including BMI, HbA1c, insulin, and HOMA-IR, although insulin was only significantly related to TSAT



A comparison of clinical characteristics between subjects with and without high iron or TSAT is shown in Table 3. Subjects with high iron and TSAT were more likely than others to drink and smoke with a slight difference. However, both high iron and TSAT were inversely associated with hsCRP and metabolic values, including BMI, HbA1c, insulin, and HOMA-IR, although insulin was only related to TSAT.

We also described reviewer’s concern in discussion section of the revised manuscripts as below;

(from line 340 to line 342 on page 26);

This gave us sufficient statistical power and could be more relevant to healthy population with normal lung functions. 



This gave us sufficient statistical power and could be more relevant to healthy population with normal lung functions. However, it should be considered that it is possible to show statistical significance for small differences in the large sample sized studies. And, our study demonstrated modest association between hyperferritinemia and lung function parameters. There also have been discordant results on this issue among studies [3-6], although those were undertaken in populations where the serum ferritin level were lower than the current study. It is suggested that a variety of factors are likely to have more influence on lung function than ferritin in the real world, and vice versa. Therefore, we cannot exclude the possibility of some unmeasured or residual confounding factors in the association between ferritin and lung function. 

C3-2.

It seems mathematically odd here that the FVC and FEV1 values shown for the combined groups match that of the smaller hyper- group rather than that of the normal group making up over 80% of the study population.

R3-2.

We believe the odd FVC and FEV1 values in table 1 might be the rounding off the numbers to three decimal places, as you can see below table. Therefore, we reported the value of FVC and FEV1 to three decimal places to decrease the mathematical add.

 All subjects (n=42,927) Normal ferritin Hyperferritinemia

 (ferritin ≤300 ng/mL) (n=34,743, 80.7%) (ferritin >300 ng/mL) (n=8,184, 19.3%)

FVC (L) † 4.73±0.56 4.75±0.55 4.73±0.56

 4.734±0.556 4.745±0.545 4.730±0.557

FEV1(L) † 3.88±0.47 3.89±0.47 3.88±0.48

 3.884±0.474 3.891±0.467 3.879±0.477

FVC% predicted† 99 ± 9 99 ± 9 98 ± 9

 98.62±8.83 98.87±8.88 98.43±8.63

FEV1% predicted† 100 ± 9 100 ± 9 99 ± 9

 99.54±9.32 99.61±9.36 99.22±9.12

C3-3.

Rounding is also an issue for bilirubin and creatinine, where the reported numbers for the two groups appear identical, but for bilirubin the star indicates a highly significant difference.

R3-3.

And we apologize for our typo. Originally, the value of total bilirubin in hyperferritinemia group was 1.0 ± 0.4 in table 1. We changed the value of total bilirubin in table 1 as bellow;

 All subjects (n=42,927) Normal ferritin Hyperferritinemia p value

 (ferritin ≤300 ng/mL) (n=34,743, 80.7%) (ferritin >300 ng/mL) (n=8,184, 19.3%) 

Total bilirubin (mg/dL) (n=47,980)* 0.9 ± 0.4 0.9 ± 0.4 0.9 ± 0.4 <0.001

Total bilirubin (mg/dL) (n=47,980)* 0.9 ± 0.4 0.9 ± 0.4 1.0 ± 0.4 <0.001

C4.

The lower portion of Table 1 shows a slight inverse relationship of the ferritin levels vs FVC and FEV1 quartiles. This is seen in both the normal and hyper ferritin groups, contrary to the idea that there may be a threshold effect for ferritin to affect lung function. It is not clear what the significance test (“compared with hyperferritinemia”) means here; is there a difference in the relationship between the two?...and if so, which is stronger? For both FVC and FEV1, the % difference in ferritin level from quartile 1 to 4 is greater for the normal group, although the absolute differences are larger in the hyper- group

R4.

We apologize for the lack of clarity. We believe that wrong presentation for significance test could be a result of confusion. Serum ferritin level was decreased across increasing quartile of FVC% (p=0.001) and FEV1% (p<0.001) in all subjects, using Kruskal-Wallis tests. So, marks such as “†” and “*” did not mean significant difference between the two groups (normal ferritin vs. hyperferritinemia). Therefore, we had better remove the lower portion of Table 1 to avoid the confusion. We really appreciate that we had a chance to revise this. Interestingly, inverse relationship of the ferritin levels across increasing quartile of FVC% and FEV1% showed statistical significance in both two groups. Therefore, it seems to be contrary to the idea that there may be a threshold effect for ferritin to affect lung function, as you pointed out. We can’s fully explain the reasons for this. However, the median and mean level of ferritin in our subjects with normal ferritin level was 177ng/ml which was still much higher than Ghio’s study (mean: 74ng/ml) [3] and Lee’s study (median: 62.3 ng/mL) [4] showing a contrary effect for ferritin on the lung function to ours. Thus, the association between ferritin (not dichotomizing ferritin) and lung function might be depend on serum ferritin level of study subjects. Consequently, we still believed threshold effect or double-edged characteristics for ferritin on lung function which could be hypothesis that might explain mixed findings and different serum ferritin level among studies. However, unfortunately the accurate threshold level is not known and there has been no consensus on criteria for hyperfrritinemia till now. Therefore, there could be debate about whether our definition of hyperferritinemia may be appropriate threshold for protection from respiratory damage. We also do not thick that our definition of hyperferritinemia may be appropriate threshold value. We just selected this level, based on the usual upper normal limits of ferritin commonly used in the literature [7] and our center. Besides, careful consideration is required when interpretating the findings from the large sample sized studies, because it is possible to show statistical significance for small differences, as you pointed earlier. Slight inverse relationship of the ferritin levels vs FVC and FEV1 quartiles in the normal ferritin group could not be clinically meaningful, although it show statistical significance. Therefore, further prospective studies are needed to determine whether ferritin is an independent risk of lung function deterioration and what the threshold level is appropriate.

We remove the lower portion of Table 1 to avoid the confusion. Instead, we add some sentences on the our definition of hyperferritinemia and the uncertainty for appropriate threshold level, to reflect your concerns

(from line 342 on page 26);

This gave us sufficient statistical power and could be more relevant to healthy population with normal lung functions. However, it should be considered that it is possible to show statistical significance for small differences in the large sample sized studies. And, our study demonstrated modest association between hyperferritinemia and lung function parameters. There also have been discordant results on this issue among studies [3-6], although those were undertaken in populations where the serum ferritin level were lower than the current study. It is suggested that a variety of factors are likely to have more influence on lung function than ferritin in the real world, and vice versa. [7-11]. Therefore, we cannot exclude the possibility of some unmeasured or residual confounding factors in the association between ferritin and lung function. (see response C3-1 which we commented above) Nonetheless, several limitations need to be addressed.



This gave us sufficient statistical power and could be more relevant to healthy population with normal lung functions. However, it should be considered that it is possible to show statistical significance for small differences in the large sample sized studies. And, our study demonstrated modest association between hyperferritinemia and lung function parameters. There also have been discordant results on this issue among studies [3-6], although those were undertaken in populations where the serum ferritin level were lower than the current study. It is suggested that a variety of factors are likely to have more influence on lung function than ferritin in the real world, and vice versa. Therefore, we cannot exclude the possibility of some unmeasured or residual confounding factors in the association between ferritin and lung function. Furthermore, there has been no consensus on criteria for hyperfrritinemia and, there could be debate about whether our definition of hyperferritinemia may be appropriate threshold for protection from respiratory damage. Even if our study with the large sample size demonstrated significant association between hyperferritinemia and lung function parameters, further prospective studies are needed to determine whether ferritin is an independent risk of lung function deterioration and what the threshold level is appropriate. Our study also had several limitations.

We removed below sentence from line 306 on page 24 to avoid overlapping.

“although the accurate threshold level is not known.”

.

C5. Table 2

The % numbers in title are confusing and should be removed. They show the distribution between groups, while the similarly displayed % values in the table do not, but are based upon the total in each group.

R5.

As you recommended, we removed the % numbers in title. And, we also removed the % numbers at title in table 3. Thank you for your advice.

And we apologize for our typo in table 3. The number of subjects with iron value was 35,890 not 34,873 

References

1. (1995) Standardization of Spirometry, 1994 Update. American Thoracic Society. Am J Respir Crit Care Med 152: 1107-1136.

2. West JB (2013) GOLD Executive Summary. Am J Respir Crit Care Med 188: 1366-1367.

3. Ghio AJ, Hilborn ED (2017) Indices of iron homeostasis correlate with airway obstruction in an NHANES III cohort. Int J Chron Obstruct Pulmon Dis 12: 2075-2084.

4. Lee CH, Goag EK, Lee SH, Chung KS, Jung JY, Park MS, et al. (2016) Association of serum ferritin levels with smoking and lung function in the Korean adult population: analysis of the fourth and fifth Korean National Health and Nutrition Examination Survey. Int J Chron Obstruct Pulmon Dis 11: 3001-3006.

5. Brigham EP, McCormack MC, Takemoto CM, Matsui EC (2015) Iron status is associated with asthma and lung function in US women. PLoS One 10: e0117545.

6. McKeever TM, Lewis SA, Smit HA, Burney P, Cassano PA, Britton J (2008) A multivariate analysis of serum nutrient levels and lung function. Respir Res 9: 67.

7. Adams PC, Barton JC (2011) A diagnostic approach to hyperferritinemia with a non-elevated transferrin saturation. J Hepatol 55: 453-458.

---

## [Decision Letter · Decision Letter 2]

5 Mar 2020

PONE-D-19-26396R2

Manuscript ID: PONE-D-19-26396

Decreased lung function is associated with elevated ferritin, but not iron or transferrin saturation in 42,927 healthy Korean men: a cross-sectional study

PLOS ONE

Dear Dr. Song,

Thank you for submitting your manuscript to PLOS ONE. After careful consideration, we feel that it has merit but does not fully meet PLOS ONE’s publication criteria as it currently stands. Therefore, we invite you to submit a revised version of the manuscript that addresses the points raised during the review process.

The reviewer still had some minor comments regarding the presentation and grammar errors.

We would appreciate receiving your revised manuscript by Apr 19 2020 11:59PM. To enhance the reproducibility of your results, we recommend that if applicable you deposit your laboratory protocols in protocols.io, where a protocol can be assigned its own identifier (DOI) such that it can be cited independently in the future. For instructions see: http://journals.plos.org/plosone/s/submission-guidelines#loc-laboratory-protocols

We look forward to receiving your revised manuscript.

Kind regards,

Yu Ru Kou, PhD

Academic Editor

PLOS ONE

Reviewers' comments:

Reviewer's Responses to Questions

**Comments to the Author**

1. If the authors have adequately addressed your comments raised in a previous round of review and you feel that this manuscript is now acceptable for publication, you may indicate that here to bypass the “Comments to the Author” section, enter your conflict of interest statement in the “Confidential to Editor” section, and submit your "Accept" recommendation.

Reviewer #2: (No Response)

2. Is the manuscript technically sound, and do the data support the conclusions?

Reviewer #2: Yes

3. Has the statistical analysis been performed appropriately and rigorously? 

Reviewer #2: I Don't Know

4. Have the authors made all data underlying the findings in their manuscript fully available?

Reviewer #2: No

5. Is the manuscript presented in an intelligible fashion and written in standard English?

Reviewer #2: Yes

6. Review Comments to the Author

Reviewer #2: Abstract

The FVC and FEV1 values are now more accurately stated in Table 1, but no change was made in the Abstract so that the rounded numbers indicate differences of 1% which is more than double the actual values of ~ 0.4%. These should be restated with the same values as in the table.

Study design

In describing the excluded subjects it would be more straightforward to state the exclusion (rather than inclusion) criteria, ie

…participants who had lung function impairment, defined as forced expiratory volume in one second (FEV1) to forced vital capacity (FVC) [FEV1/FVC] < 0.7 and FVC < 80% predicted [19], …

Lung Function measurement

The deletion of the equations, requires proofreading for appropriate changes in the accompanying text, ie

delete “the following” and change : to .

Results

3rd and 4th sentences are awkwardly stated. Suggest:

Comparison of clinical variables between the two groups showed small, but significant, differences in age, …..

Compared with the normal ferritin group, subjects in the hyperferritinemia group had lower values of spirometry….

Discussion

p13 line 154 typo hyperferritinemia

7. PLOS authors have the option to publish the peer review history of their article (what does this mean?). If published, this will include your full peer review and any attached files.

Reviewer #2: No

---

## [Author Response · Author response to Decision Letter 2]

10 Mar 2020

C1. Abstract

The FVC and FEV1 values are now more accurately stated in Table 1, but no change was made in the Abstract so that the rounded numbers indicate differences of 1% which is more than double the actual values of ~ 0.4%. These should be restated with the same values as in the table.

R1.

We do appreciate the reviewer’s comment. We apologize for our carelessness.

As reviewer pointed out, we restated with the values of predicted FVC and FEV1 to three decimal places as in the table 1, to avoid the confusion.

(from line 39 to line 40 on page 2 );

Subjects with hyperferritinemia had lower FEV1% and FVC% than those with normal ferritin level (99% vs.100% for FEV1%, p = 0.015 and 98% vs. 99% for FVC, p = 0.001).



Subjects with hyperferritinemia had lower FEV1% and FVC% than those with normal ferritin level with a slight difference, but those were statistically significant (99.22% vs.99.61% for FEV1%, p = 0.015 and 98.43% vs. 98.87% for FVC, p = 0.001).

C2. Study design

In describing the excluded subjects it would be more straightforward to state the exclusion (rather than inclusion) criteria, ie 

…participants who had lung function impairment, defined as forced expiratory volume in one second (FEV1) to forced vital capacity (FVC) [FEV1/FVC] < 0.7 and FVC < 80% predicted.

R2-A

We do appreciate the reviewer’s comment.

As reviewer pointed out, we stated the exclusion criteria rather than inclusion criteria in the manuscript as follows;

(from line 86 to line 89 on page 4 );

We excluded 85,455 participants who had an impaired lung function (the subjects without normal lung function defined as forced expiratory volume in one second (FEV1) to forced vital capacity (FVC) [FEV1/FVC] ≥ 0.7 and FVC ≥80% predicted) [1]



We excluded 85,455 participants who had a ventilation disorder (pure restriction: forced expiratory volume in one second (FEV1) to forced vital capacity (FVC) [FEV1/FVC] ≥ 0.7 and FVC < 80% predicted; pure obstruction: FEV1/FVC < 0.7 and FVC ≥80% predicted and mixed ventilation disorder: FEV1/FVC < 0.7 and FVC < 80% predicted) [1] 

C3. Lung Function measurement

The deletion of the equations, requires proofreading for appropriate changes in the accompanying text, ie

delete “the following” and change : to

R3.

We apologize for our carelessness. Thank you for pointing our mistakes.

As you recommended, we removed “the following” and modified as follows;

And we modified reference because reference article was published officially.

(J Korean Med Sci. 2019 Dec 9; 34(47):e304)

(from line 139 to line 141 on page 6 );

The predicted values for FEV1 and FVC were calculated from the following equations obtained in a representative Korean population sample [2];



The predicted values for FEV1 and FVC were calculated from equations to obtain in a representative Korean population sample [2].

C4. Results

3rd and 4th sentences are awkwardly stated. Suggest:

Comparison of clinical variables between the two groups showed small, but significant, differences in age, …..

Compared with the normal ferritin group, subjects in the hyperferritinemia group had lower values of spirometry….

R4.

Thank you for your advice. We apologize for our awkward presentation.

As you recommended, we modified sentences as follows;

(from line 169 to line 173 on page 7 );

When compared clinical variables between two groups, small but significant differences were seen in age, smoking habit, alcohol intake, liver function, CRP, blood pressure and a variety of metabolic parameters, including BMI, fasting glucose, and HbA1c. Compared with the normal ferritin group, subjects in the hyperferritinemia group were to have lower value of spirometry with a narrow margin, although those were statistically significant.



Comparison of clinical variables between the two groups showed small, but significant difference in age, smoking habit, alcohol intake, liver function, CRP, blood pressure and a variety of metabolic parameters, including BMI, fasting glucose, and HbA1c. Compared with the normal ferritin group, subjects in the hyperferritinemia group had lower values of spirometry with a narrow margin, although those were statistically significant.

C5. Discussion

p13 line 154 typo hyperferritinemia

R5.

We apologize for our carelessness and the lack of clarity Thank you for pointing our typo out. 

We changed “hyperfrritinemia” into “hyperferritinemia” in discussion section (from line 348 on page 24 ).

---

## [Decision Letter · Decision Letter 3]

16 Mar 2020

Manuscript ID: PONE-D-19-26396

Decreased lung function is associated with elevated ferritin, but not iron or transferrin saturation in 42,927 healthy Korean men: a cross-sectional study

PONE-D-19-26396R3

Dear Dr. Song,

We are pleased to inform you that your manuscript has been judged scientifically suitable for publication and will be formally accepted for publication once it complies with all outstanding technical requirements.

With kind regards,

Yu Ru Kou, PhD

Academic Editor

PLOS ONE

Additional Editor Comments (optional):

Reviewers' comments:

Reviewer's Responses to Questions

**Comments to the Author**

1. If the authors have adequately addressed your comments raised in a previous round of review and you feel that this manuscript is now acceptable for publication, you may indicate that here to bypass the “Comments to the Author” section, enter your conflict of interest statement in the “Confidential to Editor” section, and submit your "Accept" recommendation.

Reviewer #2: All comments have been addressed

2. Is the manuscript technically sound, and do the data support the conclusions?

Reviewer #2: Yes

3. Has the statistical analysis been performed appropriately and rigorously? 

Reviewer #2: I Don't Know

4. Have the authors made all data underlying the findings in their manuscript fully available?

Reviewer #2: No

5. Is the manuscript presented in an intelligible fashion and written in standard English?

Reviewer #2: Yes

6. Review Comments to the Author

Reviewer #2: Thank you for the revisions and corrections. I have no additional comments.

I have nothing more so say but must type 100 characters!@%$

7. PLOS authors have the option to publish the peer review history of their article (what does this mean?). If published, this will include your full peer review and any attached files.

Reviewer #2: No

---

## [Editor Report · Acceptance letter]

18 Mar 2020

PONE-D-19-26396R3 

Decreased lung function is associated with elevated ferritin but not iron or transferrin saturation in 42,927 healthy Korean men: A cross-sectional study 

Dear Dr. Song:

I am pleased to inform you that your manuscript has been deemed suitable for publication in PLOS ONE. Congratulations! Your manuscript is now with our production department. 

With kind regards,

on behalf of

Dr. Yu Ru Kou 

Academic Editor

PLOS ONE